# Drivers and determinants of extreme humanitarian needs among Rohingya refugee households: Evidence from UNHCR's multi-sectoral needs analysis

Harry Wilson [ORCID]*

School of Public Health and Preventive Medicine, Monash University, Melbourne, Victoria, Australia

* harry.j.wilson@unswalumni.com

## Abstract

### Background

The Rohingya refugee crisis continues to deteriorate amid major funding cuts and a myriad of intensifying threats. Approximately one million Rohingya refugees are currently housed within 33 densely populated camps in Cox's Bazar, Bangladesh. This study aimed to estimate the prevalence of households in extreme humanitarian need, identify sector-specific drivers, and elucidate household characteristics associated with extreme humanitarian needs to strategically inform humanitarian relief efforts.

### Methods

Data were sourced from the 2023 Joint Multi-Sectoral Needs Assessment (J-MSNA) – a representative cross-sectional survey of Rohingya refugee households. Households were selected via stratified simple random sampling from UNHCR's database of registered refugees. Data were collected from 3,400 households and 18,172 household members distributed across the 33 camps of Cox's Bazar through face-to-face interviews using a pre-tested structured questionnaire between August and September 2023. Survey-adjusted logistic regression was used to elucidate household characteristics associated with extreme humanitarian needs.

### Results

A total of 1,206 households (35.5%) were found to be in extreme humanitarian need, predominantly driven by sector-specific needs within the education (17.2%), food security (12.7%), and health sectors (7.2%). In adjusted analysis, household characteristics significantly associated with increased odds of extreme humanitarian needs included female head of households (aOR = 1.65, 95% CI = 1.29–2.12), head of household age between 30–49 years (aOR = 1.58, 95% CI = 1.27–1.96), or age 50

**Data availability statement:** The de-identified participant-level dataset underlying the study results may be available upon reasonable request to UNHCR at https://microdata.unhcr.org/index.php/catalog/1128 subject to approval based on their data sharing policies, ethical requirements, and data use agreements.

**Funding:** The author(s) received no specific funding for this work.

**Competing interests:** The authors have declared that no competing interests exist.

and older (aOR = 2.18, 95% CI = 1.74–2.72), increasing household size (aOR = 1.20, 95% CI = 1.15–1.25), and households with at least one member experiencing symptoms of psychosocial distress or trauma (aOR = 1.25, 95% CI = 1.06–1.46).

## Conclusion

The findings attest to the deteriorating Rohingya refugee crisis in Cox's Bazar. The household characteristics associated with extreme humanitarian needs highlight the repercussions of contemporary funding cuts among vulnerable cohorts that pay the heaviest price. The resurgence of targeted violence in Myanmar and intensity of the protracted crisis in Bangladesh demands a more compassionate and enduring humanitarian response from the international community.

## Introduction

Almost eight years since the mass displacement of Myanmar's Rohingya ethnic minority in late 2017, the protracted humanitarian crisis is rapidly deteriorating and the delivery of life saving relief services are endangered by dwindling financial support from the international community [1–4]. Approximately one million Rohingya refugees currently reside among the 33 densely populated camps located in Cox's Bazar – Bangladesh. Recent assessments indicate a myriad of intensifying threats including the rising prevalence of malnutrition associated with substantial cuts to food rations, major landslides that have destroyed thousands of refugee shelters, and increased hospitalizations influenced by concurrent outbreaks of Dengue, Rubella, Measles and suspected Chikungunya [1–3,5–8]. The socioeconomic vulnerability of aid-dependent refugee households has reached a dangerous precipice. While resurgent violence in Myanmar continues to prohibit the safe, dignified and voluntary repatriation of Rohingya refugees.

This study utilises data collected from Rohingya refugee households during the 2023 Joint Multi-Sectoral Needs Assessment (J-MSNA). The J-MSNA was coordinated by the Inter-Sector Coordination Group (ISCG) with funding from the United Nations High Commissioner for Refugees (UNHCR), Directorate-General for European Civil Protection and Humanitarian Aid Operations (ECHO), and the International Organization for Migration (IOM) [9,10]. The J-MSNA acts as a foundational monitoring framework that contextually informs the humanitarian response plan for Rohingya refugee households within the 33 camps of Cox's Bazar [9–11]. Building upon prior assessments conducted since 2018, the 2023 J-MSNA marks the fifth and latest survey among Rohingya refugee households [9–15].

Utilising the 2023 J-MSNA data, this study aimed to estimate the prevalence of households experiencing extreme humanitarian needs, identify sector-specific living standard gaps that drive extreme household needs, and elucidate characteristics associated with households in extreme need to strategically inform targeted humanitarian relief efforts.

## Methods

### Study design and setting

This study utilised de-identified participant level data from the 2023 J-MSNA, a representative cross-sectional survey of registered Rohingya refugee households located within the 33 camps of Cox's Bazar [10]. The primary data collection was conducted by the REACH Initiative with technical support from sector specific partners [16,17]. The dataset was accessed on the 27th of March 2025 under license approved by UNHCR.

### Participants and sampling strategy

The study population encompassed all registered Rohingya refugee households within the 33 camps of Cox's Bazar [16,17]. An exhaustive sampling frame of household addresses was compiled for each camp by UNHCR and IOM [16,17]. Households were selected for assessment using a stratified (by camp) simple random sampling technique [16,17]. The number of households sampled from each camp was informed by calculating the maximum sample size required to achieve a 95% confidence level and 10% margin of error to ensure the study was sufficiently powered [16,17]. The sample size was adjusted for the finite household population at each camp, and inflated by 10% to mitigate anticipated non-response [16,17]. A total of 3465 households were approached to participate, complete data (without missing fields) were collected from 3,400 households (response rate >98%) including 18,172 individual household members were surveyed from the 33 refugee camps between August 27th and September 17th, 2023 [16,17].

### Data collection

Participant data were collected through face-to-face interviews using a pre-tested structured questionnaire that captured standardised humanitarian indicators and sector specific response items [16,17]. Trained gender-balanced (male-female) field research teams conducted the household interviews with an adult (18 + years) respondent of the corresponding gender (self-identified) [16,17]. Participant self-reported responses were recorded on mobile tablets utilising the Kobo Collect platform and regularly uploaded to a secure UNHCR database for processing [16,17]. Data were routinely cleaned according to the REACH Initiative's minimum data quality standards [16,17]. A final anonymized dataset with all identifiable information removed was generated for independent analysis.

### Variables

The primary outcome was households in extreme humanitarian need defined as the presence of at-least one extreme living standard gap across the six key humanitarian sectors. The primary outcome aligns with the common "people in need (PiN)" key humanitarian indicator, adapted to the household setting, and restricted to extreme severity scores as per past J-MSNA vulnerability assessments due to outcome homogeneity using the conventional severe (severity score = 3) or extreme (severity score = 4+) cut-off points [15,18]. Sector-specific extreme needs were assigned to households according to the attainment of sector-specific criteria that have been contextually adapted from the REACH Initiative's Multi-Sectoral Needs Index (MSNI), prior J-MSNA assessments conducted among the Rohingya refugee camps, and updated operational guidelines established by the Joint Intersectoral Analysis Framework (JIAF) [9–15,18]. Household characteristics evaluated for association with extreme humanitarian needs were identified from previous J-MSNA, Refugee Influx Emergency Vulnerability Assessments (REVA), and funding impact analysis conducted by the World Food Programme [8–15,19].

### Statistical analysis

All results were calculated using Stata version 19.5 survey functions that accounted for the stratified sampling design, adjusted for the clustering of individuals within the same household, and controlled for differential probability of household selection between the camps (strata) using survey weights calculated as the inverse probability of selection.

The age-gender structure of sampled household members was contrasted against the UNHCR database of registered Rohingya refugees (sampling frame) to assess sample representativeness. Descriptive statistics were computed to analyse the distribution of household and individual characteristics. Weighted frequencies (n) and percentages (%) were reported for categorical variables. Median and interquartile range (IQR) were reported for asymmetrically distributed continuous variables. Mean and standard deviation (SD) were reported for symmetrically distributed continuous variables. Prevalence estimates were reported as weighted percentages (%) with 95% confidence intervals (95% CIs) for precision. The prevalence of households in extreme humanitarian need was disaggregated by sector and indicator. The number of concurrent sectoral needs and pairwise correlation between sector-specific needs was conducted to inform the validity of the primary outcome.

Bivariate and multivariate logistic regression analysis was conducted to investigate the association of household characteristics with the odds of extreme humanitarian needs. Unadjusted (OR) and adjusted odds ratios (aOR) were reported for bivariate and multivariate estimates respectively. A threshold value of $p \leq 0.05$ was used to conclude statistical significance and 95% CIs were reported for precision. Head of household age was treated as a categorical variable using appropriate cut-off points identified from the outcome probability distribution across head of household age (Fig 1A in S1 Fig). Household size was evaluated for linear effect and consequentially treated as a continuous variable (Fig 1B in S1 Fig). Effect modification was investigated between head of household gender and marital status using interaction terms and stratified analyses to evaluate distortion of main effects concentrated among single women. Pairwise correlation and variance inflation factor (VIF) analysis was conducted to evaluate potential problematic multicollinearity. Sensitivity analyses was conducted to evaluate measurement bias due to overlapping criteria between household characteristics used as both predictors and indicators of extreme humanitarian needs. The logistic regression models were re-fitted and odds-ratios re-estimated for head of household gender after excluding gender-specific health indicators (unmet needs for antenatal/postnatal care, safe delivery, or gender-based violence services), increasing household size after excluding overcrowding criteria (>3 persons per room), and households with at least one member experiencing psychosocial distress or trauma after excluding unmet mental health and psychosocial support services (MHPSS) within the health sector.

### Ethics

This study utilised de-identified participant-level data curated by UNHCR for independent analysis. As no additional interaction with human participants was conducted, no additional ethical approval was required in alignment with other research articles that have used J-MSNA data [20]. The ethical requirements of the primary data collection were appropriately addressed by ISCG and the REACH Initiative. Verbal informed consent was obtained from all household respondents and documented in the secondary dataset used for analysis [16,17]. Participants were informed of the study's purpose, the voluntary nature of participation, confidentiality of their responses, and their right to skip any question or withdraw from the interview at any time without consequence [16,17]. Enumerators were extensively educated on the principals of respondent safeguarding and trained to provide appropriate referral pathways upon discovery of urgent protection or health issues [16,17].

## Results

### Participant characteristics

The age-gender profile of sampled household members (N = 18,172) strongly aligns with the UNHCR database of registered Rohingya refugees (Fig 1). The median age among household members was 16 years (IQR = 7–29; Table 1), and over half were children under the age of 18 (53.3%; Fig 1). The gender distribution of household respondents (N = 3,400) interviewed by enumerators was approximately balanced with a slight underrepresentation of females (47.5%; Table 1). The median age of household respondents was 34 years (IQR = 26.0–48.0) and half of all respondents were aged

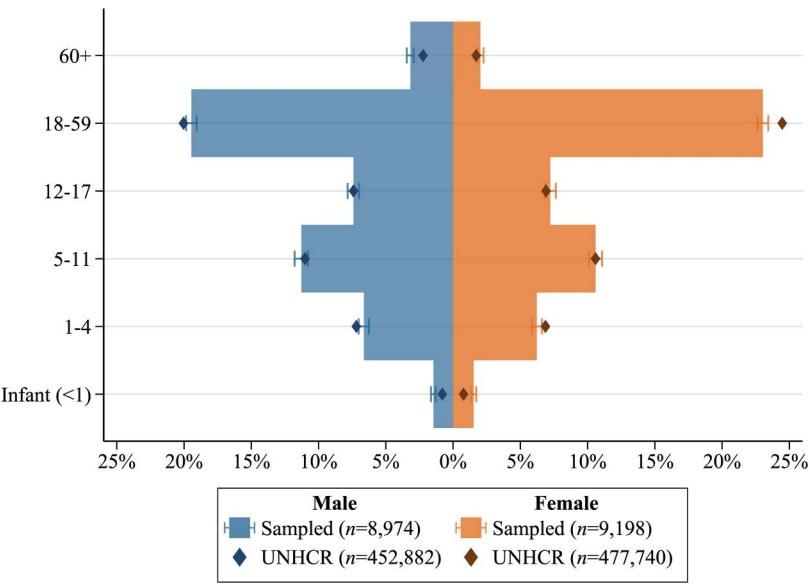

**Fig 1. Age-gender structure of sampled household members (% [95% CI]) relative to UNHCR database of registered Rohingya refugees (sampling frame).**

between 18–34 years old (50.3%; Table 1). Approximately 17.1% (95% CI = 15.8%–18.6%) of households included at least one member living with a disability, and 38% (95% CI = 36.3%–39.7%) included at least one member experiencing symptoms of psychosocial distress or trauma in the past two weeks. The median monthly household income from all sources was US$17.2 (IQR = $12.4–$24.8; Table 1) per capita, well below the US$25.2 minimum expenditure basket (MEB), and predominantly contributed by humanitarian aid [7].

## Extreme humanitarian needs

Of the 3,400 households sampled, 1,206 (35.5%, 95% CI = 33.8%–37.2%; Table 2) were found to be in extreme humanitarian need, defined as the presence of at-least one extreme living standard gap across the six key humanitarian sectors. The distribution of households in need varied substantially by sector and were most prevalent within the education (17.2%, 95% CI = 15.9%–18.6%), food security (12.7%, 95% CI = 11.6%–13.9%), and health sectors (7.2%, 95% CI = 6.3%–8.2%); Table 2).

Subpopulation analysis of school-age children (5–18 years; n = 7,204) revealed that 10.9% (95% CI = 10.0%–11.8%: Supplementary Table 1A in S1 Table) are in extreme educational need, due to serious child protection risks while at or travelling to school (3.2%, 95% CI = 2.7%–3.8%), working to earn income instead or required at home to support the household (6.0%, 95% CI = 5.3%–6.7%), and underage marriage or early pregnancy (1.7%, 95% CI = 1.3%–2.1%). Disaggregation by gender further elucidated that the prevalence of educational needs was approximately 2.5-fold higher (OR=2.47, 95% CI = 2.06–2.96, p < 0.001) among school-age girls (15.1%, 95% CI = 13.7%–16.6%) relative to school-age boys (6.7%, 95% CI = 5.8%–7.7%; Supplementary Table 1B in S1 Table).

Over a quarter of sampled households (27.3%; S2 Table) were experiencing needs in a single humanitarian sector, an additional 6.4% were in need across two humanitarian sectors, and 1.4% of households were experiencing extreme needs across three or more sectors (S2 Table). The strongest correlation of sector-specific needs was observed between the education and food security sectors (r = 0.117). All other pairwise combinations of sector-specific extreme needs demonstrated weak correlations (r < 0.10).

**Table 1. Characteristics of sampled households (N = 3,400) and household members (N = 18,172).**

| | Summary*<br>N = 3,400 |
|---|---|
| **Age, years (Median [IQR]) (N = 18,172)[a]** | 16.0 [7.0–29.0] |
| Male (median age [IQR]) (n = 8,974) | 16.0 [7.0–30.0] |
| Female (median age [IQR]) (n = 9,198) | 17.0 [7.0–29.0] |
| **Gender (N = 18,172)[a]** | |
| Male | 8,974 (49.4%) |
| Female | 9,198 (50.6%) |
| **Respondent Age, y (Median [IQR])** | 34.0 [26.0–48.0] |
| 18-34 | 1,711 (50.3%) |
| 35-59 | 1,261 (37.1%) |
| 60+ | 428 (12.6%) |
| **Respondent Gender** | |
| Male | 1,786 (52.5%) |
| Female | 1614 (47.5%) |
| **Year of Arrival in Bangladesh** | |
| Before 2016 | 360 (10.6%) |
| During 2016 | 75 (2.2%) |
| During 2017 | 2,941 (86.5%) |
| After 2017 | 24 (0.7%) |
| **Household Prevalence** | |
| At least one Member Living with a Disability[b] | 17.1% [15.8%–18.6%] |
| At least one Member Experiencing Psychosocial Distress or Trauma[c] | 38.0% [36.3%–39.7%] |
| **Household Size** | |
| Small (1–4) | 1,286 (37.8%) |
| Medium (5–7) | 1,562 (46.0%) |
| Large (8+) | 552 (16.2%) |
| **Household Monthly Income (2023 US$ per Capita)[d]** | |
| All Sources (Median [IQR]) | $17.2 [$12.4–$24.8] |
| Independent Sources (Median [IQR]) | $4.5 [$0.0–$10.2] |

*Estimates are relative to the total number of sampled households (N = 3,400) unless otherwise stated. Summary statistics are frequency (%) for categorical variables, or median [IQR] for continuous variables.

[a]Estimate relative to the sample size of all household members (N = 18,172).

[b]Disability defined as per WG-SS guidelines: "a lot of difficulty" or "cannot do it all" to any of the six functional domains (seeing, hearing, walking or climbing stairs, remembering or concentrating, self-care, and communication) [21].

[c]Psychosocial distress or trauma defined as experiencing any of the following symptoms: Nightmares, lasting sadness, extreme fatigue without doing work, often tearful; hopeless for the future; avoiding people, places or activities due to feelings of distress; anxiety or extreme worry for the future; extreme anger and out of control; uninterested in things that they used to like; unable to carry out essential activities; changes in appetite or sleep pattern compared to usual.

[d]Exchange rate used by the World Food Programme as of October 2023, 1 USD = 110 BDT [7].

## Household characteristics associated with extreme humanitarian needs

When unadjusted for the effects of other household characteristics in bivariate analysis, extreme humanitarian needs were significantly associated with female headed households (OR = 1.28, 95% CI = 1.07–1.52; Table 3), head of household age

**Table 2. Households in extreme humanitarian need by sector and indicator.**

| Sector | n (% [95% CI])* |
|---|---|
| **Protection:** | **8 (0.2% [0.1%–0.5%])** |
| *(i)* At least one child (<18) is not living with the household due to reasons that indicate an extreme protection threat (arbitrary detention, missing, kidnapped/abducted, or engagement with armed groups/forces) | 8 (0.2% [0.1%–0.5%]) |
| **Education:** | **585 (17.2% [15.9%–18.6%])** |
| *(i)* At least one school aged child (5–18) is learning in an unsafe environment that presents an extreme safety threat (protection risks whilst at or travelling to school, gender-based violence, or risk of being recruited by armed forces/groups/gangs) | 169 (5.0% [4.2%–5.8%]) |
| *(ii)* At least one school aged child (5–18) is not enrolled or regularly attending school due to extreme circumstances (protection risks whilst at or travelling to school, working to afford essential needs, childhood marriage/pregnancy, or recruited by armed forces/groups/gangs) | 432 (12.7% [11.6%–13.9%]) |
| **Health:** | **245 (7.2% [6.3%–8.2%])** |
| *(i)* At least one household member is living with a disability[a] and reported an unmet healthcare need. | 84 (2.5% [2.0%–3.1%]) |
| *(ii)* At least one household member reported an unmet healthcare need for a major clinical reason (trauma, chronic illness, emergency/lifesaving surgery, antenatal or postnatal support, safe delivery, gender-based violence, or mental health and psychosocial support) | 211 (6.2% [5.4%–7.1%]) |
| **Water, Sanitation, and Hygiene (WASH):** | **55 (1.6% [1.3%–2.1%])** |
| *(i)* Household is using an unimproved source of drinking water (unprotected well or unprotected spring)[b] | 0 (0.0%) |
| *(ii)* Household frequently experienced a shortage of drinking water (≥11 times per month) | 55 (1.6% [1.2%–2.1%]) |
| *(iii)* Household is using an unimproved sanitation facility (flush/pour flush to open drain, pit latrine without slab/open pit, bucket/plastic bag, hanging toilet/hanging latrine, and no facility/bush/field)[b] | 1 (<0.1% [0.0%–0.2%]) |
| **Food Security:** | **431 (12.7% [11.6%–13.9%])** |
| *(i)* Household food consumption categorised as phase 4 or 5 according to the Integrated Food Security Phase Classification (IPC) indicating an acute emergency or catastrophe[c] | 3 (0.1% [0.0%–0.3%]) |
| *(ii)* Household utilised or exhausted emergency livelihood coping strategies for essential needs in the past 30 days (children under 18 years working to contribute to household income, high risk illegal income activities, socially degrading activities such as begging and/or scavenging to meet essential needs, marriage of children under 18 years due to financial pressure or to secure a more sustainable life among the host population)[d] | 430 (12.6% [11.5%–13.9%]) |
| **Shelter & Non-Food Items:** | **194 (5.7% [4.9%–6.6%])** |
| *(i)* Total collapse of household's primary shelter | 44 (1.3% [1.0%–1.8%]) |
| *(ii)* Household is severely overcrowded with more than three people per room | 155 (4.5% [3.9%–5.3%]) |
| **Total Households in Extreme Humanitarian Need:** | **1,206 (35.5% [33.8%–37.2%])** |

*Percentage estimates are relative to the total number of sampled households (N = 3,400). Note that sector-specific estimates are not perfectly additive as households may satisfy one or more indicators.

[a]Disability classified as per Washington Group Short Set (WG-SS) guidelines: Respondent cited "a lot of difficulty" or "cannot do it at all" to any of the six functional domains (seeing, hearing, walking or climbing stairs, remembering or concentrating, self-care, and communication) [21].

[b]Drinking water sources and sanitation facilities were categorised according to the WHO/UNICEF Joint Monitoring Programme (JMP) [22].

[c]IPC classification assigned to the household according to the Famine Early Warning Systems Network (FEWS NET) standard matrix criteria using Food Consumption Score (FCS), Reduced Coping Strategies Index (rCSI), and Household Hunger Scale (HHS) sub-indicators (See S3 Table).

[d]Emergency coping strategies defined as per the World Food Programme Livelihood Coping Strategies Indicator for Food Security [23].

between 30–49 years (OR = 2.18, 95% CI = 1.78–2.68), head of household age 50 and older (OR = 2.73, 95% CI = 2.20–3.39), increasing household size (OR = 1.20, 95% CI = 1.15–1.24), and households with at least one member experiencing symptoms of psychosocial distress or trauma (OR = 1.33, 95% CI = 1.15–1.55). These characteristics all demonstrated independent associations after adjustment in multivariate analysis with similar effect sizes. Notably, female headed households were more strongly associated (aOR = 1.65, 95% CI = 1.29–2.12), whereas head of household age between 30–49 years (aOR = 1.58, 95% CI = 1.27–1.96), or age 50 and older (aOR = 2.18, 95% CI = 1.74–2.72) were less strongly associated after adjustment for other household characteristics. No evidence of problematic multicollinearity was observed,

**Table 3. Household characteristics associated with the odds of extreme humanitarian needs.**

| | Summary* | Unadjusted | | Adjusted | |
|---|---|---|---|---|---|
| | N = 3,400 | OR (95% CI) | p-value | aOR (95% CI) | p-value |
| **Head of Household Gender** | | | | | |
| Male | 2,671 (78.5%) | 1.00 | – | 1.00 | – |
| Female | 729 (21.5%) | **1.28 (1.07-1.52)** | **0.007*** | **1.65 (1.29-2.12)** | **<0.001*** |
| **Head of Household Age, years** | | | | | |
| 18-29 | 837 (24.6%) | 1.00 | – | 1.00 | – |
| 30-49 | 1,480 (43.5%) | **2.18 (1.78-2.68)** | **<0.001*** | **1.58 (1.27-1.96)** | **<0.001*** |
| 50+ | 1,083 (31.9%) | **2.73 (2.20-3.39)** | **<0.001*** | **2.18 (1.74-2.72)** | **<0.001*** |
| **Head of Household Marital Status** | | | | | |
| Married | 2,929 (86.1%) | 1.00 | – | 1.00 | – |
| Single, Separated, Divorced, or Widowed | 471 (13.9%) | 1.04 (0.84-1.29) | 0.691 | 0.88 (0.66-1.19) | 0.411 |
| **Monthly Household Income (Excluding Aid)** | | | | | |
| No Income (Aid Dependent) | 991 (29.1%) | 1.00 | – | 1.00 | – |
| Income Below Monthly Expenditure Basket for Food Needs[a] | 2,073 (61.0%) | 0.98 (0.83-1.16) | 0.780 | 0.88 (0.73-1.06) | 0.191 |
| Income Equal to or Above Monthly Expenditure Basket for Food Needs[a] | 336 (9.9%) | 0.76 (0.57-1.00) | 0.053 | 0.98 (0.73-1.31) | 0.882 |
| **Household Size** | 5 (2) | **1.20 (1.15-1.24)** | **<0.001*** | **1.20 (1.15-1.25)** | **<0.001*** |
| **At Least One Member Experiencing Psychosocial Distress or Trauma[c]** | | | | | |
| No | 2,108 (62.0%) | 1.00 | – | 1.00 | – |
| Yes | 1,292 (38.0%) | **1.33 (1.15-1.55)** | **<0.001*** | **1.25 (1.06-1.46)** | **0.006*** |

*Summary statistics are frequency (%) for categorical variables or mean (sd) for household size.

[a]Monthly expenditure basket for food needs represents the average income required for a household to meet their basic food needs ($17.60 USD per capita per month) [7].

[b]Psychosocial distress or trauma defined as experiencing any of the following symptoms: nightmares, lasting sadness, extreme fatigue without doing work, often tearful; hopeless for the future; avoiding people, places or activities due to feelings of distress; anxiety or extreme worry for the future; extreme anger and out of control; uninterested in things that they used to like; unable to carry out essential activities; changes in appetite or sleep pattern compared to usual.

all pairwise correlation coefficients of independent variables were below 0.7 and the mean VIF in the multivariable model was 2.6 with no extreme outliers (VIFmax < 10). No evidence of an interaction between head of household gender and marital status was observed with stratified analysis confirming consistent main effects among married and single female headed households. Sensitivity analysis demonstrated that female headed households (aOR = 1.68, 95% CI = 1.32–2.15; p < 0.001), households with at least one member in psychosocial distress (aOR = 1.24, 95% CI = 1.06–1.45; p = 0.008), and increasing household size (aOR = 1.16, 95% CI = 1.12–1.21; p < 0.001) were independently associated with extreme humanitarian needs when decoupled from overlapping indicator criteria (S4 Table).

## Discussion

Despite immense humanitarian efforts since 2017, over one-third of Rohingya refugee households were found to be in extreme humanitarian need. The high prevalence of households in extreme need aligns with findings from prior J-MSNA and empirically supports field observations from relief agencies that bear witnessed to the deterioration of key humanitarian sectors [2–4,6,7,19]. The strong alignment between the age-gender profile of sampled households and the UNHCR database of registered Rohingya refugees living within the 33 camps vigorously supports the representativeness of participants and the generalisability of our results. The rigorous sampling method used to select households and face-to-face interviews of respondents is of particular resonance when contrasted against prior J-MSNA conducted in 2020 and 2021 [14,15]. These iterations collected data via phone interviews due to COVID-19 restrictions, consequentially leading to

sampling bias towards males (>75% of all respondents) and more affluent households with higher rates of phone ownership [14,15].

The sector-specific analysis highlighted the education, food-security and health sectors as the predominate drivers of extreme humanitarian needs experienced by households. The utilisation of emergency livelihood-based coping strategies behind the high prevalence of extreme food security needs strongly aligns with independent analysis conducted by the World Food Programme that further evidence the impacts of successive food ration cuts [7,19].

Concerningly, almost half of all households in extreme humanitarian need were attributed to the education sector that necessitates additional mitigation efforts. The sector-specific analysis highlights the need for a more complex multisectoral response that redresses both the serious protection threats that endangers the safety of school-age children and the broader survival pressures that provoke households to deprioritise their child's education as a negative coping strategy. Ostensibly, these dynamics exemplify the education sector's dual role as an independent driver and symptom of humanitarian needs. Future research should seek to strategically inform tailored intervention through elucidating the age-specific and gender-specific educational needs of school-age children. This is further reinforced by the 2.5-fold prevalence of extreme educational needs observed among school-age girls that quintessentially captures both the gendered burden and generational impact of the protracted humanitarian crisis. Irrespective of the need for further evidence, these tailored interventions remain undeliverable without commensurate funding to resource them. These challenges also undermine multisectoral efforts that are required to adequately address interdependencies in other sectors through improving the broader living conditions within the refugee camps.

The household characteristics associated with extreme humanitarian needs complement and quantify prior vulnerability assessments [8,15]. The logistic regression analysis demonstrated that female-headed households, older heads of household, larger households, and households including members experiencing psychosocial distress or trauma were independently associated with increased odds of extreme humanitarian needs. These associations exemplify the disproportionate impact shouldered by households with greater vulnerability and reduced adaptive capacity.

Our findings are subject to numerous strengths and limitations. Selection bias was minimized through stratified simple random sampling within each camp and the use of survey weights that controlled for differential probabilities of household selection between the camps. The strong alignment between the age-gender structure of the sample and the UNHCR database suggested minimal sampling bias. Confounding was addressed through multivariable logistic regression that revealed minor confounding between head of household age and gender. Potential effect modification was appropriately investigated between head of household gender and marital status using stratified analysis that confirmed effect homogeneity. Potential measurement bias due to criterion overlap was addressed through sensitivity analysis that decoupled predictive variables from their overlapping indicators. The cross-sectional design precludes causal inferences. The composite outcome does not distinguish the severity of households experiencing extreme needs in a single sector from those with needs across multiple sectors. However, the sector-specific analysis demonstrated that households experiencing extreme multisectoral needs was rare and significantly influenced by shared criteria between sectors. The weak correlation between sector-specific outcomes further validates the composite outcome's sensitivity to detect different domains of humanitarian needs across key sectors and best informs the targeted delivery of humanitarian relief by local authorities.

## Conclusion

The findings attest to the deteriorating humanitarian situation of the Rohingya refugee response in Bangladesh [2–4,6,7,19]. More than a third of Rohingya refugee households were found to be in extreme humanitarian need. Household characteristics associated with extreme humanitarian needs included female headed households, older head of households, increasing household size and households with at-least one member experiencing symptoms of psychosocial distress or trauma. These findings emphasize the profound gravity of contemporary funding cuts that disproportionately impact more vulnerable households that pay the heaviest price. The resurgence of violence

in Myanmar nullifies short-term hopes of repatriation and further exacerbates the funding requirements needed to deliver an effective humanitarian response. The scale of humanitarian needs among Rohingya refugee households paired with the proliferating demands for humanitarian aid provoked by concurrent global crisis presents a challenging crucible for the international community to unequivocally affirm the human rights of forcibly displaced peoples both in Bangladesh and abroad [6].

## Supporting information

**S1 Table. Prevalence of extreme education needs among school aged children disaggregated by reason and gender.**
(DOCX)

**S2 Table. Distribution of households in extreme humanitarian need by number and combination of sectoral needs.**
(DOCX)

**S3 Table. FEWS NET food security matrix results.**
(DOCX)

**S4 Table. Sensitivity analysis of household characteristics associated with the odds of extreme humanitarian needs after removing overlapping indicator criteria.**
(DOCX)

**S1 Fig. Effect linearity analysis of extreme humanitarian needs by head of household age and household size.**
(DOCX)

## Acknowledgments

The author wishes to formally acknowledge those whose collective efforts have made this research article possible. Specifically, the Inter-Sector Coordination Group (ISCG), UNHCR, IOM, ECHO, the REACH Initiative, enumerators, other study personnel, and household respondents that graciously donated their time.

## Author contributions

**Conceptualization:** Harry Wilson.

**Data curation:** Harry Wilson.

**Formal analysis:** Harry Wilson.

**Funding acquisition:** Harry Wilson.

**Investigation:** Harry Wilson.

**Methodology:** Harry Wilson.

**Project administration:** Harry Wilson.

**Resources:** Harry Wilson.

**Software:** Harry Wilson.

**Supervision:** Harry Wilson.

**Validation:** Harry Wilson.

**Visualization:** Harry Wilson.

**Writing – original draft:** Harry Wilson.

**Writing – review & editing:** Harry Wilson.

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
