## [Decision Letter · Decision Letter 0]

27 Oct 2025

Dear Dr. Wilson,

Thank you for submitting your manuscript to PLOS ONE. After careful consideration, we feel that it has merit but does not fully meet PLOS ONE’s publication criteria as it currently stands. Therefore, we invite you to submit a revised version of the manuscript that addresses the points raised during the review process.

We look forward to receiving your revised manuscript.

Kind regards,

Jennifer Yourkavitch

Academic Editor

PLOS ONE

Journal Requirements:

3. For studies involving third-party data, we encourage authors to share any data specific to their analyses that they can legally distribute. PLOS recognizes, however, that authors may be using third-party data they do not have the rights to share. When third-party data cannot be publicly shared, authors must provide all information necessary for interested researchers to apply to gain access to the data. (https://journals.plos.org/plosone/s/data-availability#loc-acceptable-data-access-restrictions)

4. We note you have included a table to which you do not refer in the text of your manuscript. Please ensure that you refer to Table 3 in your text; if accepted, production will need this reference to link the reader to the Table.

Additional Editor Comments :

Please respond to each comment. Noting here that Reviewer #2 is the Academic Editor.

Reviewers' comments:

Reviewer's Responses to Questions

**Comments to the Author**

1. Is the manuscript technically sound, and do the data support the conclusions?

Reviewer #1: Yes

Reviewer #2: Yes

2. Has the statistical analysis been performed appropriately and rigorously?

Reviewer #1: Yes

Reviewer #2: Yes

3. Have the authors made all data underlying the findings in their manuscript fully available?

Reviewer #1: Yes

Reviewer #2: Yes

4. Is the manuscript presented in an intelligible fashion and written in standard English?

Reviewer #1: Yes

Reviewer #2: Yes

Reviewer #1: On page 4, lines 75-76, pls clarify the 95% confidence level used, but the margin of error was set to 10%? Should it not be 5%, at 95% confidence level?

On page 7, lines 176-177 - i suggest reiterating how extreme humanitarian need is defined. While this was mentioned in the methods section, it helps to mention again in the results to guide readers.

Given that education is the top driver of humanitarian need, i wonder why households with school-going age population not included as a predictor? The paper did not offer any recommendation, moving forward. How do we address the education need of the young in humanitarian setting? I encourage the authors to reflect on this.

Reviewer #2: 1. “Data” is a plural term. Please check subject – verb agreement throughout the manuscript.

2. Add study design to title or abstract

3. Describe potential sources of bias and any efforts to address them

4. What was the refusal rate for participating in the survey? Consider a flow diagram to show how many approached, how many eligible, how many responded, how man included in the analysis?

5. Were there missing data? How was missing data addressed?

6. Construct the conclusion to reflect what can be concluded from this study. Other commentary belongs in the Discussion section.

**Do you want your identity to be public for this peer review?** For information about this choice, including consent withdrawal, please see our Privacy Policy

Reviewer #1: No

Reviewer #2: No

---

## [Author Response · Author response to Decision Letter 1]

29 Oct 2025

Dear Editor and Reviewers,

Thank you for your generous contributions, I sincerely appreciate the time you have spent to review my research manuscript.

I have revised the manuscript and submitted these files as per the recommendations.

Please see my responses to the review feedback below:

Editor

See markup for changes made and files renamed.

The ethics statement is within the methods section. A slight alteration was made to move the data access date into the study design and setting section of the method.

3. For studies involving third-party data, we encourage authors to share any data specific to their analyses that they can legally distribute. PLOS recognizes, however, that authors may be using third-party data they do not have the rights to share. When third-party data cannot be publicly shared, authors must provide all information necessary for interested researchers to apply to gain access to the data. (https://journals.plos.org/plosone/s/data-availability#loc-acceptable-data-access-restrictions)

The data availability statement has been amended to include the specific URL. This data source is also explicitly referenced in the methods section (see reference 10 in manuscript).

4. We note you have included a table to which you do not refer in the text of your manuscript. Please ensure that you refer to Table 3 in your text; if accepted, production will need this reference to link the reader to the Table.

Table 3 has been referred to in-text after the estimates provided for the first variable in that section. See manuscript markup.

Not applicable

All citations have been validated and formatted as per requirements

Reviewer 1

1. On page 4, lines 75-76, pls clarify the 95% confidence level used, but the margin of error was set to 10%? Should it not be 5%, at 95% confidence level?

This is based on the sampling strategy designed by the external parties that conduced the primary data collection. The confidence interval (z-score of 1.96) and margin of error (plus or minus 10%) are different components used to calculated the sample size rather than the reciprocal of each other.

The primary objective of the original data collection was to inform camp-specific gaps for a wide range of outcomes to monitor the response. The use of 10% margin of error in the sample size calculation was used to ensure that 95% of the time the camp-specific result had a minimum level of precision (+/- 10%) irrespective of the prevalence within the specific camp.

The margin of error is also different to the alpha threshold (5%) used to conclude statistical significance during hypothesis testing.

2. On page 7, lines 176-177 - i suggest reiterating how extreme humanitarian need is defined. While this was mentioned in the methods section, it helps to mention again in the results to guide readers.

I respectfully disagree and do not believe that restating how the outcome was defined is required in both the methods and results section. This aspect is more relevant to the variables portion of the methods section where it has been made available if the reader needs to scroll back and re-read. I think this change would also disrupt the logical flow of the manuscript.

3. Given that education is the top driver of humanitarian need, i wonder why households with school-going age population not included as a predictor? The paper did not offer any recommendation, moving forward. How do we address the education need of the young in humanitarian setting? I encourage the authors to reflect on this.

This variable would likely induce the measurement bias from criterion overlap between this predictor (school age children) and the education related indicators (At least one school aged child not attending/not enrolled or learning in an unsafe environment for extreme reasons) used to assign the outcome (see Reviewer 2: Response 3 for more details).

This exact variable was considered (households with school-age children) but rejected due to homogeneity as it categorised almost all households. A similar linear variable (number of school age children within the household) was considered, as were other variables relating to dependents (school age children, those with a disability or the elderly). These were all too highly correlated with increasing household size, which was selected in preference to mitigate criterion overlap and due to superiority in a mutually adjusted model. In essence, household size was actually a better predictor than the number of school age children or another similar variable related to the household composition.

Regarding recommendations for the education sector, this is an excellent point and exactly where the research is headed. The results prompt a much more granular sector-specific analysis of the education sector. What may seem like a surface level acknowledgment is actually a lead-in to a prospective paper that is underway. This work will inform specific educational interventions that require more careful elucidation by age-group and gender. These interventions are not deliverable without adequate funding. Additionally, in the context of this paper regarding broader humanitarian needs, I have addressed the underlying dynamics of the educational sector in relation to its role as both an independent driver and symptom of the living conditions that exacerbate negative household coping strategies. From this perspective, redressing the educational needs is inextricably linked to addressing the household’s humanitarian needs and improving the living conditions within the camps.

Reviewer 2

1. “Data” is a plural term. Please check subject – verb agreement throughout the manuscript.

See markup for changes where required.

2. Add study design to title or abstract

No changes were made as the study design was stated in the first sentence of the abstract’s methods section:

“Data was sourced from the 2023 Joint Multi-Sectoral Needs Assessment (J-MSNA) - a representative cross-sectional survey of Rohingya refugee households”

3. Describe potential sources of bias and any efforts to address them

I believe that these efforts are inherently addressed by the sampling design that was described and statistical analysis that was conducted without the need for a revision in the manuscript to accommodate an exhaustively list of bias. For clarity, specific sources of bias were addressed as follows:

Stratified simple random sampling and survey weights were used to control for selection bias (ie. random selection of households within each camp and survey weights to address the differential probabilities of household selection between the camps).

The age-gender structure of the sample was contrasted against the population to evidence any potential sampling bias. No such bias was evidence that otherwise require acknowledgment or further statistical remediation.

Multivariable logistic regression to control for confounding. Particularly the age and gender structure demonstrated minor confounding after adjustment which is specifically highlighted in the results section.

Potential effect heterogeneity (effect modification) was specifically investigated between marital status and head of household gender. This was based on the odds associated with female headed households that may be due to concentrated effects among single women. As per the results section, stratified analysis was used to confirm effect homogeneity in this particular circumstance.

The criterion overlap is another structural issue due to household characteristics being used as both indicators (such as severe overcrowded) and predictors (such as increasing household size) of the outcome (households in extreme humanitarian need). This issue was addressed through two techniques. Firstly, distancing the predictive variables from the indicator variables by using more general household characteristics rather than those explicitly used as logic to assign the outcome. Secondly, sensitivity analysis was conducted using all indicators except those with the potential to overlap. This confirmed the presence of an independent association and provided a counterfactual to contrast the effect sizes reported in the main analysis against to ensure no problematic distortion arose from criterion overlap.

4. What was the refusal rate for participating in the survey? Consider a flow diagram to show how many approached, how many eligible, how many responded, how man included in the analysis?

The response rate was >98% (3400 households sampled, 3465 households invited).

Given the scope of results presented, I do not believe an additional figure is required or would add additional depth to the method. A research terms of reference document and methodological overview that covers these queries are publicly available and referenced in-text (see references 16 and 17 of the manuscript).

Instead of a flow diagram, an individual level age-gender diagram of sampled participants versus the population registered in UNHCRs database was provided (Figure 1). I believe has more utility through informing the generalisability of the results than extending the methods section which is described in adequate details with further details available elsewhere for those inclined. I am happy to address any specific queries from readers through correspondence.

5. Were there missing data? How was missing data addressed?

No data was missing and therefore no acknowledgement of missing data was provided.

6. Construct the conclusion to reflect what can be concluded from this study. Other commentary belongs in the Discussion section.

I agree and have toned down the statements made in the conclusion. The updated text recites the main findings, applies their implications within the research context of the challenges facing the humanitarian response and provides a call to actions for the international community to ensure that the response is adequately funded as per the humanitarian needs.

Regards,

Harry J Wilson (B.BMed.Sci, MPH, MIDI, MGH)

---

## [Editor Report · Decision Letter 1]

2 Nov 2025

Dear Dr. Wilson,

We look forward to receiving your revised manuscript.

Kind regards,

Jennifer Yourkavitch

Academic Editor

PLOS ONE

Journal Requirements:

Additional Editor Comments:

Thank you for this revision and for your responses to reviewers' comments. Since you chose to rebut most comments rather than address them, we must return to you requesting revisions again.

1. Line 175: "...were found to be in extreme humanitarian need, defined as the presence of at-least one extreme living standard gap across the six key humanitarian sectors." Adding the comma and definition as requested by the reviewer adds clarity. A reader should not have to search through your paper to remember how you chose to define that variable.

2. Regarding recommendations for the education sector, returning to Reviewer #1 comment: "The paper

did not offer any recommendation, moving forward. How do we address the education

need of the young in humanitarian setting? I encourage the authors to reflect on this." You offer thoughts in your response. Please include a statement about how this paper "addressed the underlying dynamics of the educational sector in relation

to its role as both an independent driver and symptom of the living conditions that

exacerbate negative household coping strategies." And the importance of programs and future research "addressing the household’s

humanitarian needs and improving the living conditions within the camps" to redress educational needs.

3. Describe the potential sources of bias and efforts to address them in the Discussion, as you did in the response to reviewers. You can summarize but this needs to be addressed. This is best practice. See the STROBE guidelines for observational studies for reference.

4. Include a statement about response rate and missing data in the text alongside the references provided (16-17).

---

## [Author Response · Author response to Decision Letter 2]

3 Nov 2025

Dear Editor,

Thank you for reviewing the revised manuscript. I have made further edits as per your requests. Please see my responses below:

1. Line 175: "...were found to be in extreme humanitarian need, defined as the presence of at-least one extreme living standard gap across the six key humanitarian sectors." Adding the comma and definition as requested by the reviewer adds clarity. A reader should not have to search through your paper to remember how you chose to define that variable.

I agree that this improves the clarity of the manuscript and have made this change.

2. Regarding recommendations for the education sector, returning to Reviewer #1 comment: "The paper did not offer any recommendation, moving forward. How do we address the education need of the young in humanitarian setting? I encourage the authors to reflect on this." You offer thoughts in your response. Please include a statement about how this paper "addressed the underlying dynamics of the educational sector in relation to its role as both an independent driver and symptom of the living conditions that exacerbate negative household coping strategies." And the importance of programs and future research "addressing the household’s humanitarian needs and improving the living conditions within the camps" to redress educational needs.

I have split the paragraph previously at line 249 to better focus and extend the discussion regarding the education sector. I have also included the recommendation for future research to inform tailored interventions that address specific educational needs. Please see the manuscript markup for details.

Please note as per the cover letter, the manuscript was provided in-advance to both UNHCR and the Inter Sector Coordination Group (ISCG) that coordinate the Rohingya refugee response before the original submission. I am hopeful that this has helped inform the targeted delivery of humanitarian relief to those who are more likely to be experiencing extreme need. However, given that almost every household is classified as experiencing humanitarian need, the intended purpose of this research was to provide a citable resource that has been peer-reviewed and advocate for appropriate funding arrangements commensurate to the scale of refugee households in humanitarian need by highlighting the vulnerable cohorts that are disproportionately affected.

Please also note that as an independent author who has volunteered to conduct this work without any funding, I am limited in the solutions I can provide to address complex issues such as childhood marriages or recruitment by armed gangs that contribute to the extreme educational needs experienced by school-age children. Regardless, I hope that my analysis sheds light on the situation and provides a valuable entry point for others to build upon.

3. Describe the potential sources of bias and efforts to address them in the Discussion, as you did in the response to reviewers. You can summarize but this needs to be addressed. This is best practice. See the STROBE guidelines for observational studies for reference.

I have made the requested revisions in the final paragraph of the discussion. Please see the markup in the revised manuscript for details.

I am happy to accommodate the desired changes but please note that “efforts to address potential sources of bias” are explicitly listed under the methods section in the STROBE guidelines. I still believe that the method section of the original manuscript adequately addressed these issues which are inextricably linked to how participants were sampled, data was collected and the statistical analysis was conducted.

The discussion section narrowed the focus to potential sources of bias that undermined the application of the results rather than discussing how sources of bias were addressed. This complemented the discussion piece that highlighted the generalisability of the key results and provided a contextual interpretation using other findings from the literature (past MSNA assessments and World Food Programme REVA).

4. Include a statement about response rate and missing data in the text alongside the references provided (16-17).

Agreed, I have made the requested revisions in the methods section. See markup lines 79-80.

Regards,

Harry J Wilson (B.BMed.Sci, MPH, MIDI, MGH)

---

## [Editor Report · Decision Letter 2]

17 Nov 2025

Drivers and Determinants of Extreme Humanitarian Needs among Rohingya Refugee Households: Evidence from UNHCR’s Multi-Sectoral Needs Analysis.

PONE-D-25-44967R2

Dear Dr. Wilson,

We’re pleased to inform you that your manuscript has been judged scientifically suitable for publication and will be formally accepted for publication once it meets all outstanding technical requirements.

Kind regards,

Jennifer Yourkavitch

Academic Editor

PLOS ONE

---

## [Editor Report · Acceptance letter]

PONE-D-25-44967R2

PLOS ONE

Dear Dr. Wilson,

I'm pleased to inform you that your manuscript has been deemed suitable for publication in PLOS ONE. Congratulations! Your manuscript is now being handed over to our production team.

Kind regards,

on behalf of

Dr. Jennifer Yourkavitch

Academic Editor

PLOS ONE